# Medically Serious Suicide Attempts in Personality Disorders

**DOI:** 10.3390/jcm10184186

**Published:** 2021-09-16

**Authors:** Marta Quesada-Franco, Luis Pintor-Pérez, Constanza Daigre, Enrique Baca-García, Josep Antoni Ramos-Quiroga, María Dolores Braquehais

**Affiliations:** 1Department of Psychiatry, Hospital Universitari Vall d’Hebron, 08035 Barcelona, Spain; cdaigre@vhebron.net (C.D.); jaramos@vhebron.net (J.A.R.-Q.); 2Department of Psychiatry and Legal Medicine, Universitat Autònoma de Barcelona, 08193 Barcelona, Spain; 3Department of Psychiatry, Hospital Clinic, Instituto de Investigaciones Biomédicas Augusto Pi I Sunyer (IDIBAPS), Universitat de Barcelona, 08036 Barcelona, Spain; LPINTOR@clinic.cat; 4Biomedical Network Research Centre on Mental Health (CIBERSAM), 08035 Barcelona, Spain; mdbraquehais.paimm@comb.cat; 5Department of Psychiatry, Madrid Autonomous University, 28017 Madrid, Spain; ebacgar2@yahoo.es; 6CIBERSAM (Centro de Investigacion en Salud Mental), Carlos III Institute of Health, 28040 Madrid, Spain; 7Department of Psychiatry, Centre Hospitalier Universitaire de Nîmes, 30900 Nîmes, France; 8Psychology Department, Universidad Católica de Maule, Talca 3605, Chile; 9Department of Psychiatry, University Hospital Jimenez Diaz Foundation, 28020 Madrid, Spain; 10Psychiatry, Mental Health and Addictions Research Group, Vall d’Hebron Research Institute, Universitat Autònoma de Barcelona, 08035 Barcelona, Spain; 11Integral Care Program for Sick Health Professionals, Galatea Clinic, 08017 Barcelona, Spain

**Keywords:** nearly lethal suicide attempts, serious suicide attempts, personality disorders

## Abstract

Medically serious suicide attempts (MSSA) represent a subgroup of clinically heterogeneous suicidal behaviors very close to suicides. Personality disorders (PD) are highly prevalent among them, together with affective and substance use disorders. However, few studies have specifically analyzed the role of PD in MSSA. These suicide attempts (SA) are usually followed by longer hospitalization periods and may result in severe physical and psychological consequences. The aim of this study is to compare the profile of MSSA patients with and without PD. MSSA were defined according to Beautrais ‘criteria, but had to remain hospitalized ≥48 h. Overall, 168 patients from two public hospitals in Barcelona were evaluated during a three-year period. Mean hospital stay was 23.68 (standard deviation (SD) = 41.14) days. Patients with PD (*n* = 69) were more likely to be younger, female, make the first and the most serious SA at a younger age, reported recent stressful life-events and more frequently had previous suicide attempts compared to those without PD. However, no differences were found with regards to comorbid diagnoses, current clinical status, features of the attempt, or their impulsivity and hopelessness scores. Therefore, focusing on the subjective, qualitative experiences related to MSSA among PD patients may increase understanding of the reasons contributing to these attempts in order to improve prevention strategies in the future.

## 1. Introduction

Suicide is a serious global public health issue. Globally, 703,000 people die by suicide every year and more than one in every 100 deaths (1.3%) in 2019 were the result of suicide [1]. Suicide occurs throughout the lifespan and is the fourth leading cause of death in young people aged 15–29 years for both sexes [1].

Individuals with a history of nearly lethal suicide attempts or medically serious suicide attempts (MSSA) are at higher risk for later suicide than those who make less severe suicide attempts [2,3]. Studies have shown that survivors of MSSA are epidemiologically very similar to those who die by suicide [4,5]. Consequently, recent research has focused on subjects with MSSA, as the study of this subgroup can best shed light on deaths by suicide [3].

The definition criteria for MSSA are heterogeneous. A simple definition is suicide attempts with serious medical injuries regardless of the severity of the attempter’s mental disorder. Other studies define MSSA according to psychometric scales such as: the Self-inflicted Injury Severity (SIISF) [6,7,8]; the Lethality Scale [9]; the Lethality of Suicide Attempt Scales II (LSARS-II) [10,11]; the Lethality Rating Scale (LRS) [3,12,13,14] or the Risk Rescue Rating Scales (RRRS) [15,16,17]. Others identify MSSA as those needing hospitalization in non-psychiatric facilities as a result of the medical consequences of the attempt [3,12,13,18].

Beautrais & collaborators have provided a specific clinical definition of MSSA: patients who require hospital admission for more than 24 h after the attempt and meet one of the following treatment criteria: (a) treatment in a specialized clinical unit (i.e., intensive care, hyperbaric or burn units); (b) need of surgery under general anesthesia; (c) need of medical treatment beyond gastric lavage, activated charcoal, and/or routine neurological observations; and (d) patients who require hospital admission for more than 24 h not because of the aforementioned [2,19,20,21,22,23,24]

One of the most reliable and potent predictors of death by suicide across the lifespan is having a previous suicide attempt (SA) [25,26,27,28]. The presence of multiple past attempts heightens the risk of lethal attempts in both adolescents [29] and adults [25,26,30,31]. More specifically, past MSSAs are significantly associated with a higher risk of dying by suicide [28].

Personality disorders (PD) are characterized by abnormal patterns of inner experiences and behaviors [32]. The estimated prevalence of PD in the general population is about 7.8% [33]. Among psychiatric outpatients, PD becomes more frequent although prevalence data varies depending definition criteria [14]. When limited to the 10 DSM-IV personality disorders defined by specified criteria, approximately one-third of patients in psychiatric treatment are estimated to have a personality disorder. The prevalence increased by about 15% to 45.5% when the residual category of personality disorder not otherwise specified was included [34].

With respect to PD subtypes, borderline PD is the most significant PD diagnosis associated with a higher risk of lifetime suicide attempts [35]. Retrospective and prospective research has found that suicide occurs in 3–10% of all borderline PD cases [36,37,38,39]. The most significant link between suicide and borderline PD may be the instances of repeated self-injury characterized by the disorder [40]. Through repeated self-injury, people with borderline PD become practiced regarding suicidal behavior, may lose their fear of suicide and become more competent in suicide methods and, as a consequence, engage in increasingly dangerous self-harm [40]. This hypothesis is compatible with the interpersonal-psychological theory which suggests that suicidal individuals develop the ability to hurt themselves, assimilating habituation to the feelings of pain and fear that accompany suicidal behaviors [40].

Over the last three decades, many studies have provided extensive and valuable information on the contribution of personality trait profiles to suicidal behaviour [41]. Although they may include valuable information on suicide attempts regardless of their severity, MSSA have not typically been presented in the literature as a specific topic needing separate evaluation and analysis. The fact that MSSA can be conceptualized as nearly lethal suicide has led some researchers to explore this specific phenomenon

Although the most prevalent psychiatric diagnoses among MSSA are affective disorders (mainly depression) [12,15,19,42,43]. Cooper-Kazaz found that nearly half of the individuals assessed for a MSSA had a personality disorder [18]. Another study found maladjusted personality traits in 24% of the sample [44].

This is the first study conducted in Spain with MSSA admitted to urban public hospitals. The main hypothesis of this study is that there may be significant socio-demographic and clinical differences between MSSA with and without PD.

## 2. Materials and Methods

Approval for the study was obtained from Vall d’Hebron University Hospital Ethics Committee: PR(SC)50/2008. All patients were informed about its purpose and terms. Signed, informed consent was obtained from all participants before joining the study, and the study was conducted in accordance with the Declaration of Helsinki principles.

### 2.1. Study Design and Participants

This cross-sectional study was conducted over a three-year period at two tertiary university-affiliated general hospitals, with all the medical and surgical specialties and services required for patients with unusually severe, complex, or uncommon health problems [45]. Both public hospitals are located in the metropolitan area of Barcelona (Spain). Vall d´Hebron University Hospital and Clinic Hospital provide medical assistance to catchment areas covering a population of 400,000 and 540,000 people, respectively. In Spain, due to the universal coverage of its public health system, the majority of patients with MSSA are initially referred to public hospitals in order to stabilize their medical situation.

The final sample consisted of 168 adults with MSSA consecutively admitted to the Emergency Care Units and later referred to medical-surgical units due to the severe consequences of their attempt.

The definition of MSSA followed all of Beautrais’ inclusion criteria [2,19,20,21,22,23,24]. In order to select more severe MSSA, the time needing hospital treatment at non-psychiatric facilities was expanded from Beautrais’ 24 to 48 or more hours. Inclusion criteria also comprised:

1. ≥18 years old;

2. Admitted via the emergency care unit after a MSSA;

3. Meeting one of the following treatment criteria:

3.1. Treatment in specialized units;

3.2. Need of surgery (superficial cuts were excluded);

3.3. Need of medical treatment beyond basic gastric lavage, activated charcoal and/or basic neurological assessment;

3.4. Highly lethal suicide methods with a high risk of fatality, especially hanging or gunshot who were hospitalized for more than 48 h, but did not meet the preceding criteria.

Patients from Clinic Hospital (*n* = 100) were included from February 2007 to August 2010 while those from Vall d’Hebron University Hospital (*n* = 68) were included from February 2010 to October 2012.

### 2.2. Procedure and Assessment

MSSA patients were identified daily from emergency room databases. In addition to medical assistance, psychiatric assessment of MSSA was always performed during hospitalization once the medical condition was sufficiently stabilized. Researchers were expert psychiatrists with extensive experience in the healthcare field and diagnoses followed DSM-IV criteria [46]. MSSA patients were not evaluated in the emergency room but at non-psychiatric care units during the first week after their medical condition was stabilized. The clinical evaluation was conducted during several visits and included not only individual assessment but also information from their relative(s) or significant other(s). This in-depth evaluation was important to identify not only acute mental health conditions but also others needing a longitudinal assessment, such as PD. The diagnosis was contrasted with the patient’s psychiatrist during hospitalizationin order to reach a strengthen inter-rater reliability.

Psychosocial, clinical and suicidiological variables related to suicide risk and protective factors [47] were recorded using an ad hoc questionnaire that included:-Social-demographics: age, sex, nationality, marital status, educational level, employment situation, and legal problems.-Early development: family dynamics, childhood and adolescent abuse (sexual/physical and parental neglect), and financial problems.-Physical and/or sexual abuse during their adulthood.-Stressful life events the year prior to the MSSA.-Current medical morbidity (not related to the MSSA).-Psychiatric history: method of current SA; presence of SA in the five years prior to the MSSA; previous psychiatry emergency room or inpatient admissions; previous outpatient psychiatric or psychological treatment; and family history of psychiatric disorders and suicide attempts.

Psychometric assessments included:
-The Beck Hopelessness Scale (BHS) [48]: it contains 20 items, each rated on a 5-point Likert scale, the sum of the scores ranging from 20 to 100.-The Barratt Impulsiveness Scale (BIS-11) [49]: it contains 30 self-report items scored between 0 to 4 (range of total score 0–120). This scale has three sub-scales: cognitive, motor, and non-planning impulsiveness.-The Beck Suicidal Intent Scale (SIS) [50]: it is a 15-item ordinal scale that is rated by summing up the scores on each item (varying from 0 to 2), and the final score ranges from 0 to 30 points. The SIS measures specifically one’s intent to die by any suicide attempt.

All patients completed the semi-structured interview but 30 (17.8%) did not finish the self-administered psychometric assessment

### 2.3. Statistical Analysis

Bivariate and multivariate analyses were executed. First, a descriptive analysis of all variables as percentages, means, and standard deviations was conducted. To compare patients with and without personality disorders, odds ratio with 95% confidence intervals were used to analyze the relationship between binary variables. Cohen’s d was calculated for quantitative variables. Then the Bonferroni correction for multiples testing was used in order to minimize the type I error. Finally, logistic regression analyses were conducted using the variables that remained statistically significant after the Bonferroni correction. In order to avoid co-linearity, related variables were not included in the multivariate analyses. The dependent variable was PD (0 = no PD and 1 = PD). A conditional entrance method was used to select variables in the model. All analyses were performed using the SPSS version 20, except for the calculation of Cohen’s d, when JASP 0.8.6.0 software was used. All statistical hypotheses were two-sided and a value of *p* < 0.05 was considered statistically significant.

## 3. Results

### 3.1. Description of the Sample

The mean age of the sample was 45 (SD = 17.2) years; women were 52.4% of the sample (*n* = 88); 20% (*n* = 34) of all patients were living alone; and 38% (*n* = 64) had only a primary-level education. With respect to significant lifetime events, 13.7% (*n* = 23) reported a history of physical abuse and 7.1% (*n* = 12) of sexual abuse during their childhood/adolescence. Significantly, 92% (*n* = 154) identified stressful life events the year before the MSSA, the majority related to serious interpersonal difficulties (73.8%, *n* = 124) and 32.1% (*n* = 54) reported work-related problems. Physical illnesses before the attempt were present in 56.5% (*n* = 95) of the sample, and these consisted of mainly cardiovascular diseases.

When evaluating their psychiatric history, 90.5% (*n* = 152) of the subjects had a previous mental disorder, 62.5% (*n* = 105) had at least one prior SA and 42.3% (*n* = 71) of the sample had required temporary hospitalization after a SA in the past. The majority (98.8%) of the sample currently met diagnostic criteria for at least one mental disorder. Overall, the most prevalent disorders were: affective (41.9%), personality (41.9%; *n* = 69) and substance use disorders (36.9%). Only three subjects (0.2%) did not have a mental disorder, while thirteen patients (7.7%) only met criteria for an adjustment disorder. With regards to Axis I diagnosis, 94.6% (*n* = 159) of the sample had at least one disorder: 51.8% (*n* = 87) met criteria for one diagnosis, 23.8% (*n* = 40) for two diagnoses; and 19.1% (*n* = 32) for three or more disorders. Comorbidity of Axis I and Axis II (PD) diagnoses was present in 40.9% (*n* = 65).

With respect to PD patients, the majority belonged to cluster B PD (84.1%; *n* = 58) and borderline PD (95%; *n* = 55) was the most frequent cluster B subtype. Five (7.2%) patients met criteria for cluster A and six (8.7%) for cluster C personality disorders.

The methods of attempted suicide most frequently used were: medication overdose (67.3%, *n* = 113), jumping from heights (23.2%, *n* = 39), poisoning (8.3%, *n* = 14) and cuts (also 8.3%). Most patients used only one method (62.5%, *n* = 105) although 31% (*n* = 52) combined two and the rest used three or more methods (6.6%, *n* = 11). Violent means (i.e., jumping from heights, hanging, stabbing, burns and cuts) were chosen by 36.3% of t the attempters. Alcohol was a factor in 26.2% (*n* = 44) of the attempts and 37.6% (*n* = 61) reported non-alcohol drug use before the MSSA. While 43.5% (*n* = 73) reported that they had previously planned the attempt, 40.5% (*n* = 68) answered that they began to consider it less than 3 h before the attempt.

### 3.2. Comparison between SSA Patients with and without Personality Disorders

With regards to sociodemographic and clinical variables, patients with PD were significantly more likely to be women, be younger, have interpersonal problems the year prior to the current MSSA, have a history of substance use disorders, have a family history of mental disorders, and met criteria of current cocaine use disorder compared to those without PD. After Bonferroni correction, only age, interpersonal problems and family mental health history remained statistically significant (Table 1).

With respect to past and current suicidal behavior, PD patients were significantly more likely to have an earlier and more severe previous attempt compared to those without PD. These differences remained statistically significant after Bonferroni correction. The Beck Hopelessness and Suicide Intent Scale global and subscales scores were similar in MSSA patients with and without a PD, while the global BIS-11mean score was higher among those with PDs (Table 2).

As shown in Table 3, the first multivariate analysis included the sociodemographic and clinical variables proven to be statistically significant after Bonferroni correction (see Table 1). Age, sex and interpersonal problems remained independently associated with an increased likelihood of suffering from PD (Nagelkerke R^2^ = 0.227; constant value 0.354, std. error = 0.016, 71.7% corrected predicted).

A second logistic regression analysis was conducted including current psychiatric comorbidity, past and current suicide behavior and scales results proven to be significantly different after Bonferroni correction (see Table 2). Only the age of the first SA remained independently associated with an increased likelihood of meeting criteria for PD when having an MSSA (Nagelkerke R^2^ = 0.215; constant value 0.334, std. error = 0.159, 58.3 corrected predicted).

## 4. Discussion

The aim of this study was to explore the differences between MSSA patients with and without PD. Despite the extensive literature on the relationship between PD and suicide [42,51,52,53], focusing on MSSA may provide a valuable insight into this specific suicidal behavior. MSSAs were defined according to more restrictive criteria (hospitalized 48 h or more after the attempt) so they could be more accurately conceptualized as nearly lethal behaviors. In fact, the mean length of hospitalization was 23.68 days due to the serious adverse consequences of the attempt. MSSA patients meeting criteria for PD were more likely to be women, younger and have a recent interpersonal lifetime stressful event relative to those medically serious suicide attempters without a PD. Both groups were almost similar in terms of current clinical variables and index suicide attempt characteristics. However, those with PD more frequently reported a history of a first and most serious suicide attempt at a younger age. The most common PD subtype was borderline personality disorder.

Suicide attempts in PD usually occur following stressful life events, and patients describe their motivation as a way to escape from a highly overwhelming situation [54]. In our sample, interpersonal problems were significantly more frequent in the PD group in contrast to other types of stressful life events. This finding could be in line with Joiner’s studies: interpersonal strains associated with emotional dysregulation are likely to contribute to feelings of disconnection and ineffectiveness, which are significant predictors of suicidal behavior [55,56].

In our sample, PD patients more frequently reported a history of more severe and early suicidal behavior. It could be hypothesized that PD patients, especially those with borderline personality disorder, may commonly engage in repeated self-harm and, through this repeated self-injury, they overcome their resistance to enduring pain, feeling more capable of suicide and thus progressively engaging in more severe self-harming behaviors [40,55,56].

The fact that both groups were similar with regards to not only to their main Axis I diagnosis but also to other relevant current clinical and socio-demographic variables points to the need for a reconsidering of the way psychiatric assessment of MSSA is generally conducted. Mental disorder diagnosis and quantitative analysis of factors related to MSSA may be insufficient to achieve a wider comprehension of this phenomenon. More information on the nature and subjective experiences related to interpersonal problems preceding suicidal attempts in PD patients is needed [38]. More complex psychological variables, such as psychic pain or psychalgia [57,58,59] should also be considered in the global assessment of MSSA.

The main limitations of this study were: (a) its cross-sectional design that limits the knowledge on risk and protective factors related to MSSA in PD; (b) the diagnoses followed DSM-IV-TR criteria [47], but were not conducted after a structured diagnostic interview; (c) the low sample size, especially in the subsample analysis; and (d) the lack of more information on other relevant variables that have recently been related to suicide such as those related to mental pain [57,59]. Qualitative analysis of MSSA may also enrich the comprehension of the subjective experiences leading to this type of behavior.

Although it could be seen as a study limitation, changing the definition of MSSA from 24 to 48 h hospitalization after the attempt had the strength of providing a more restrictive criterion related to the medical severity of the attempt. Regrettably, no data on suicide attempts needing more than 24 h hospitalization after admission was available to address this issue in order to compare our findings to those of Beautrais and colleagues [2,19,20,21,22,23,24].

Another weakness of the study was the fact that patients hospitalized after an MSSA were frequently in a suboptimal physical and psychological condition to complete a thorough psychiatric assessment. Therefore, in our study, interest in collecting data was balanced with respect to the patient’s personal needs in those particular circumstances. These concerns limited the intensity of the clinical assessment both with respect to the length of the interviews and the information obtained after the interview. Despite this limitation, in situ evaluation of patients after a MSSA may provide a valuable insight into the factors and reasons behind their behavior. New ways of exploring the mental states leading to severe attempts should be incorporated in future research. Follow-up studies may also shed some light on how these mind states evolve once patients have nearly died by suicide.

## 5. Conclusions

A few differences can be found when comparing MSSA patients with and without PD. In this study, those with PD were more likely to be women, younger and have recent interpersonal adversities compared to those without a PD. They were also more likely to have previous and more severe SA at a younger age. Research on the role of PD in MSSA should be expanded from a categorical and dimensional approach to a more subjective, qualitative one in order to ascertain the reasons behind these findings. This knowledge may help prevent the serious medical and psychological consequences of these types of suicide attempts and, potentially, deaths by suicide.

## Figures and Tables

**Table 1 jcm-10-04186-t001:** Comparison of sociodemographic and clinical variables between SSA patients with and without personality disorders.

	Personality Disorders (*n* = 69)		NO Personality Disorders (*n* = 99)		Statistics		
	*n* or Mean	% or SD	*n* or Mean	% or SD	Cohen’s d	OR (95% CI)	*p*-Value
Sociodemographic variables Gender							
Male	26	37.7	54	54.5		0.50 (0.67–0.94)	
Age	39.19	14.20	50.42	17.68	0.687 *		*p* < 0.01 *
Immigrant	9	13.2	13	13.3		0.99 (0.40–2.48)	
Living alone	15	21.7	19	19.2		1.17 (0.55–2.50)	
High educational qualifications	50	72.5	50	51		2.53 (1.31–4.89)	
Current incomes	51	76.1	90	94.7		0.18 (0.06–0.51)	
sexual abuse *(childhood/adolescence)*	9	14.1	3	3.4		4.69 (1.22–18.09)	
Stressful life events the year prior to the index attempt							
Interpersonal problems	62	89.9	62	62.6		5.29 (2.19–12.76) *	
Economic problems	26	37.7	27	27.3		1.61 (0.83–3.11)	
Legal problems	13	18.8	11	11.1		1.86 (0.79–4.43)	
Working problems	27	39.1	27	27.3		1.71 (0.89–3.30)	
Clinical variables							
Personal mental health history	64	92.8	86	86.9		1.94 (0.66–5.70)	
Substance Use Disorders	27	39.1	23	23.2		2.12 (1.09–4.16)	
Psychotic Disorders	1	1.4	25	25.3		0.04 (0.01–0.33)	
Affective Disorders	27	39.1	39	39.4		0.99 (0.53–1.86)	
Adjustment Disorders	10	14.5	9	9.1		1.69 (0.65–4.42)	
Family mental health history	49	74.2	49	52.1		2.65 (1.33–5.25) *	
Chronic medical condition	32	46.4	57	57.6		0.64 (0.34–1.18)	
Acute medical condition	3	4.3	14	14.1		0.28 (0.08–1.00)	
Discharge							
Further hospitalization	20	29.4	41	42.3		0.57 (0.29–1.10)	
Last year inpatient treatment ^a^	21	30.4	25	25.3		1.29 (0.65–2.57)	
Last year outpatient treatment ^a^	52	75.4	63	63.6		1.75 (0.88–3.46)	
Current diagnosis							
Substance use disorders	30	43.5	31	31.3		1.69 (0.89–3.19)	
Alcohol use disorders	20	29.0	23	23.2		1.35 (0.67–2.71)	
Cocaine use disorders	12	17.4	6	6.1		3.26 (1.16–9.18)	
Cannabis use disorders	9	13.0	7	7.1		1.97 (0.69–5.58)	
Opioids use disorders	1	1.4	1	1.0		1.44 (0.09–23.44)	
Anxiolytic use disorders	10	14.5	7	7.1		2.23 (0.80–6.18)	
Psychotic disorders	1	1.4	25	25.3		0.04 (0.01–0.33)	
Affective disorders	24	34.8	45	45.5		0.64 (0.34–1.21)	
Adjustment disorders	21	30.4	26	26.3		1.23 (0.62–2.43)	
Length of hospital stay (days)	21.96	32.04	21.09	26.93		0.830	0.034

^a^: Need of psychiatric treatment; * Statistically significant after Bonferroni Correction (0.001). SSA: serious suicide attempt; OR: Odds ratio; SD: Standard deviation, CI: Confidence Interval.

**Table 2 jcm-10-04186-t002:** Differences in past and current suicide behavior between groups and in Impulsivity, Hopelessness and Suicide Intent scales.

	Personality Disorders (*n* = 69)		NO Personality Disorders (*n* = 99)		Statistics		
History of Suicide	*n* or Mean	% or SD	*n* or Mean	% or SD	Cohen’s d	*OR* (95% CI)	*p*–Value
Family history suicide behavior	20	30.3	26	26.5		1.20 (0.60–2.40)	
Previous suicide attempts	52	75.4	54	54.5		2.55 (1.30–5.01)	
Age of most serious SA	38.12	14.64	49.42	18.07	0.675		*p* < 0.001 *
Age of first SA	29.72	14.97	45.01	19.51	0.861		*p* < 0.001 *
Number of previous SA	2.94	3.20	1.42	2.57	0.533		*p* < 0.001 *
Current serious suicide attempt							
Violent means	24	34.8	37	37.4		0.89 (0.47–1.69)	
Medication overdose	48	69.6	65	65.7		1.96 (0.62–2.31)	
Poisoning	6	8.7	8	8.1		1.08 (0.36–3.28)	
Jumping from heights	18	26.1	21	21.2		1.31 (0.64–2.69)	
Hanging	1	1.4	1	1.0		1.44 (0.09–23.44)	
Cuts	5	7.2	9	9.1		0.78 (0.25–2.44)	
Stabbing	1	1.4	3	3.0		0.47 (0.05–4.62)	
Burns	2	2.9	3	3.0		0.95 (0.16–5.87)	
Drug use	23	33.3	29	29.3		1.21 (0.62–2.34)	
Barratt Impulsiveness Scale (BIS-11): Total score	47.00	14.83	40.58	16.13	0.412		*p* < 0.05
Beck Suicide Intent Scale (SIS)	16.53	6.71	16.22	6.44	0.047	0.776	
Beck Hopelessness Scale	24.46	3.28	23.60	3.39	−0.258	0.163	

SA: Suicide Attempt; * Statistically significant after Bonferroni Correction (0.01).

**Table 3 jcm-10-04186-t003:** Results of logistic regression analysis of MSSA with PD.

	Wald	df	Sig.	Exp(B)	95% CI
Model 1: Sociodemographic and clinical variables					
Age	13.6	1	0.000	1.0	0.9–1.0
Sex	5.5	1	0.019	0.4	0.2–0.9
Interpersonal problems	11.8	1	0.001	5.4	2.1–14.2
Model 2: Past and present suicidal behavior					
Age of first SA	21.2	1	0.000	0.949	0.928–0.970

Df: Degrees of freedom; Sig: statistical significance; MSSA: medically serious suicide attempts; PD: personality disorders.

## Data Availability

Data sharing is not allowed.

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
