# Peer review of "Medically Serious Suicide Attempts in Personality Disorders"

_jcm, 2021, doi:10.3390/jcm10184186_

Round 1

Reviewer 1 Report

Summary

This study investigates the prevalence and correlates of personality disorders in N=168 individuals with a medical serious suicide attempt. The authors report that those with personality disorders tended to be younger, female, and have a history of suicide attempts that started at a younger age and were more severe than those without personality disorders, and had more severe interpersonal problems just prior to the attempt. The primary problem with this study is that the authors frame their results as an exploration of factors related to suicide attempts, but without a comparison group of individuals who did not attempt suicide it is impossible to make these claims. All this data can tell us it that individuals with MSSAs and PDs are female, younger, and have a history of suicidality, something which is true of personality disorders in general. Perhaps the only unique finding is that individuals with personality disorders were more likely to report interpersonal problems than non-PD attempters, but the significance or limitations of this finding is not explored other than to suggest that more research needs to be done on psychalgia. Additional methodological limitations also weaken this manuscript.

Major Comments:

  • The authors don’t really define MSSA, either in the introduction where it would help frame their narrative or in the methods where they simply refer to “Beautrais’ criteria”. Moreover, they altered Beautrais’ criteria to only include individuals hospitalized for 48, not 24. MSSAs should be defined conceptually in the introduction and the full criteria listed in the methods, including a sufficient justification for why the extension from 24 to 48 hours was appropriate. If any data are available on those who were hospitalized between 24 and 48 hours, this should be presented to determine whether a significant subset of attempters was excluded by this rule.
  • The manuscript has a number of dubious and/or unsupported claims that undermine the narrative.
    • For instance, no citation is provided for the claim that each suicide has roughly 20 matching attempts (Lines 45-46).
    • The authors also claim that only a few studies have investigated personality disorders in MSSA (Lines 71-72); personality disorders have long been investigated in suicidality, and the fact that these studies don’t explicitly focus on “MSSA” is not a reason to ignore them.
    • Similarly, the authors claim that personality disorders are common (lines 60-61) and cite a study of outpatients where 45% had a personality disorder. According to the National Institutes of Mental Health, personality disorders affect roughly 9% of the populace, which when divided among the various PDs doesn’t make them particularly “common”.
  • There is a methodological limitation in that the measures used are largely non-validated, including a diagnosis of personality disorder that didn’t involve a structured clinical interview. Personality disorders are not easy to diagnose, and simply using DSM-IV criteria does not mean the diagnosis will be accurate, particularly if the assessment occurs shortly after an extremely traumatic event like an MSSA. The authors address their desire to not overburden patients while hospitalized, which is laudable, but the reasonable solution is to conduct a follow-up evaluation some time after the admission, not conduct a quick and dirty evaluation in the emergency department that could very likely be wrong.

Minor Comments: (take them or leave them)

  • Some English language editing would be useful.
  • Please limit the use of acronyms where possible, and make sure any acronyms are introduced properly. For instance, I don’t know what ISA means (methods line 124) because it hasn’t been defined by this point.
  • I think it would be useful to specify the personality disorders beyond “cluster B”.
  • I would give p-values their own column in Tables 1 and 2 rather than combining them with odds ratios.
  • What is the reference suicide attempt method in table 2? I’m assuming the odds ratios refer to a reference level, but it’s not clear what that is.
  • Table 3 needs some work – what is gl? Why Exp(B) and not OR? A better way would be to show all the variables included in each model with Odds ratios and 95% confidence intervals so we can see which variables were included and which ones ended up being significant.

Author Response

Dear Reviewer,

We thank you for your thoughtful and constructive comments on our manuscript. We hope the text has been improved by your suggestions.

Summary:

This study investigates the prevalence and correlates of personality disorders in N=168 individuals with a medical serious suicide attempt. The authors report that those with personality disorders tended to be younger, female, and have a history of suicide attempts that started at a younger age and were more severe than those without personality disorders, and had more severe interpersonal problems just prior to the attempt. The primary problem with this study is that the authors frame their results as an exploration of factors related to suicide attempts, but without a comparison group of individuals who did not attempt suicide it is impossible to make these claims. All this data can tell us it that individuals with MSSAs and PDs are female, younger, and have a history of suicidality, something which is true of personality disorders in general. Perhaps the only unique finding is that individuals with personality disorders were more likely to report interpersonal problems than non-PD attempters, but the significance or limitations of this finding is not explored other than to suggest that more research needs to be done on psychalgia. Additional methodological limitations also weaken this manuscript.

Response 1: We thank the reviewer for their thoughtful comments and have rewritten the abstract and the manuscript following the reviewer’s comments and concerns.

Major Comments:

  • Point 2: The authors don’t really define MSSA, either in the introduction where it would help frame their narrative or in the methods where they simply refer to “Beautrais’ criteria”. Moreover, they altered Beautrais’ criteria to only include individuals hospitalized for 48, not 24. MSSAs should be defined conceptually in the introduction and the full criteria listed in the methods, including a sufficient justification for why the extension from 24 to 48 hours was appropriate. If any data are available on those who were hospitalized between 24 and 48 hours, this should be presented to determine whether a significant subset of attempters was excluded by this rule.

Response 2: We have included some comments on the problem of MSSA definition in the introduction section and gone more in depth with respect to Beautrais’ definition. In the methods section, we have clarified the definition criteria in the Methods Section and explained the reasons behind extending the time lapse from 24 to 48 hours. We have kept the inclusion criteria explained in the first version of the manuscript. Regretfully, we had no data about suicide attempts meeting Beautrais criteria of 24 or more hours and we have included this limitation in the Discussion section.

  • Point 3: The manuscript has a number of dubious and/or unsupported claims that undermine the narrative.

Response 3: We have gone through the manuscript and have supported all claims with the corresponding evidence-based reference/s.

  • Point 4: For instance, no citation is provided for the claim that each suicide has roughly 20 matching attempts (Lines 45-46).

Response 4: We have rewritten this part of introduction and included the updated information and corresponding reference to support this claim.

  • Point 5: The authors also claim that only a few studies have investigated personality disorders in MSSA (Lines 71-72); personality disorders have long been investigated in suicidality, and the fact that these studies don’t explicitly focus on “MSSA” is not a reason to ignore them.

Response 5: We have rewritten this point, following the reviewer’s advice, and have explained that although, over the last three decades, many studies have provided extensive and valuable information on the contribution of personality trait profiles to suicidal behavior, medically serious suicide attempts have not typically been presented in the literature as a specific topic needing separate evaluation and analysis (see Introduction section).

Point 6: Similarly, the authors claim that personality disorders are common (lines 60-61) and cite a study of outpatients where 45% had a personality disorder. According to the National Institutes of Mental Health, personality disorders affect roughly 9% of the populace, which when divided among the various PDs doesn’t make them particularly “common”.

Response 6: We have clarified this point including updated information on estimated prevalence of PD in the general population and also provided data on PD’s prevalence among psychiatric outpatients although it is true that PD’s prevalence variability may be related to the breadth of PD’s definition. We have also included the corresponding updated references for both claims (see Introduction section).

  • Point 7: There is a methodological limitation in that the measures used are largely non-validated, including a diagnosis of personality disorder that didn’t involve a structured clinical interview. Personality disorders are not easy to diagnose, and simply using DSM-IV criteria does not mean the diagnosis will be accurate, particularly if the assessment occurs shortly after an extremely traumatic event like an MSSA. The authors address their desire to not overburden patients while hospitalized, which is laudable, but the reasonable solution is to conduct a follow-up evaluation some time after the admission, not conduct a quick and dirty evaluation in the emergency department that could very likely be wrong.

Response 7: We have expanded the information on how the clinical diagnosis was conducted (see Methods section). Diagnosis was conducted by expert psychiatrists with extensive experience in the healthcare field and followed DSM-IV criteria. Patients were not evaluated in the emergency room but when hospitalized in non-psychiatric care units during the first week after their medical condition was stabilized. The clinical evaluation was conducted in several visits and included not only individual assessment but also information from their relative(s) or significant other(s). This was important not only for acute mental health conditions but also for others, such as PD, that need a longitudinal assessment. The diagnosis was contrasted with the psychiatrist’s in charge of the patient during the hospitalization in order to reach a high inter-rater reliability.

Nevertheless, we are aware of this important methodological limitation and have discussed it in the Discussion section.  

Minor Comments

  • Point 8: Some English language editing would be useful.

Response 8: We thank you for the suggestion. The new version of the manuscript has been thoroughly reviewed by a native English speaking editor.

  • Point 9: Please limit the use of acronyms where possible, and make sure any acronyms are introduced properly. For instance, I don’t know what ISA means (methods line 124) because it hasn’t been defined by this point.

Response 9: In order to limit the use of acronyms, we have only used them for medically serious suicide attempts (MSSA) and for personality disorders (PDs).

  • Point 10: I think it would be useful to specify the personality disorders beyond “cluster B”.

Response 10: We have included the data on cluster A and cluster C personality disorders prevalence and provided the frequency of each cluster among PD patients.

Point 11: I would give p-values their own column in Tables 1 and 2 rather than combining them with odds ratios.

Response 11: We included the p-values both in Table 1 and 2.

  • Point 12: What is the reference suicide attempt method in table 2? I’m assuming the odds ratios refer to a reference level, but it’s not clear what that is.

Response 12: With regards to this point, the reference level is for each suicide method condition (as a dichotomic variable) method type (yes/no).

  • Point 13: Table 3 needs some work – what is gl? Why Exp(B) and not OR? A better way would be to show all the variables included in each model with Odds ratios and 95% confidence intervals so we can see which variables were included and which ones ended up being significant.

Response 13: We apologize as “gl” is the Spanish equivalent to “df” (degrees of freedom). We used Exp(B) in the regression logistic analysis as it is the equivalent to the OR in a logistic regression model and also provided the corresponding 95% Confidence Interval. We have corrected the Table 3 accordingly.

Reviewer 2 Report

Thank you for the opportunity to review the manuscript titled 'Medically Serious Suicide Attempts in Personality Disorders'. I believe this paper is of relevance to the special issue and is interesting. However, I have made some suggestions for how the manuscript could be improved.

Abstract

  • Minor comments – should be ‘focused’ not focus and the reliance on the word ‘them’ as opposed to more person-first types of language (this applies throughout).

Introduction

  • There are no sources cited for the opening two sentences of the introduction, this is problematic as the authors state close to 800,000 people die by suicide as has been reported by earlier WHO reports; however according to most recent WHO estimates which supersede earlier figures this is closer to 700,000 (the authors cite these latter 2019 estimates in the following sentence)
  • While I appreciate the authors brevity, I believe further detail on personality disorders and suicidality would be beneficial to readers and the special issue (e.g., impacts on functioning, compare prevalence in MSSA & psychiatric outpatients currently reported with general population?). There needs to be stronger rationale for this study.

Methods

  • Is there any information as to the training/experience or role of researchers conducting the assessment?
  • Mental disorders were assigned using DSM-IV-TR criteria, later it is noted as a limitation that this was not using a standardized diagnostic interview or schedule, perhaps the authors could briefly describe process and whether any inter-rater agreement was achieved?
  • How long did this assessment take with this vulnerable group? Were breaks allowed, how were these conducted during the week post MSSA? Was there any missing data?

Results

  • Could the n be provided for those who did not have current mental health disorder (these are presumably in the no PD group)
  • As expected, there is some comorbidity in the groups, is it possible to provide range in number of multiple diagnoses? Is it possible to compare average number across groups?
  • Inconsistent boldface in Table 1

Discussion

  • Overall, I feel there is a need for more ‘discussion’ of these cross-sectional findings.
  • As it stands, I am not fully grasping the link that “Our findings point to the need of including more complex psychological variables”. While I agree with the authors assertion, I feel the rationale for this statement needs better explaining.
  • The second paragraph stands out, how does this relate to current findings? Was there a prevalence of BPD in the current sample?

Conclusion

  • This section appears weak to me. “This better knowledge may help prevent the serious medical and psychological consequences of this type of suicide attempt[s] and, potentially, deaths by suicide among them.” Is this referring to the suggestions for future research or the current findings? I believe a stronger conclusion of the relevance and implications of the current study are warranted.

Minor comments

  • Some editing for English sentence tense required throughout, sometimes MSSA is incorrectly abbreviated to MMSA

Author Response

Dear Reviewer,

We thank you for your thoughtful and constructive suggestions on our manuscript. We hope the text has improved after considering your comments.

Abstract

  • Point 1:

Minor comments – should be ‘focused’ not focus and the reliance on the word ‘them’ as opposed to more person-first types of language (this applies throughout).

Response 1: We have corrected this and other errors and the new version of the manuscript has been thoroughly reviewed by a native English-speaking editor.

Introduction

  • Point 2: There are no sources cited for the opening two sentences of the introduction, this is problematic as the authors state close to 800,000 people die by suicide as has been reported by earlier WHO reports; however according to most recent WHO estimates which supersede earlier figures this is closer to 700,000 (the authors cite these latter 2019 estimates in the following sentence)

Response 2: Thank you for pointing this out. We have rewritten this part of introduction and included the updated information and corresponding reference to support this claim.

  • Point 3: While I appreciate the authors brevity, I believe further detail on personality disorders and suicidality would be beneficial to readers and the special issue (e.g., impacts on functioning, compare prevalence in MSSA & psychiatric outpatients currently reported with general population?). There needs to be stronger rationale for this study.

Response 3: We have expanded the information on the relationship between PD and suicidal behavior with an emphasis on its role on clearly defined MSSA (see Introduction section).

Methods

  • Point 4: Is there any information as to the training/experience or role of researchers conducting the assessment?

Response 4: We have expanded the information on how the clinical assessment was conducted (see Methods section). Diagnosis was conducted by expert psychiatrists with extensive experience in the healthcare field and followed DSM-IV criteria.

  • Point 5: Mental disorders were assigned using DSM-IV-TR criteria, later it is noted as a limitation that this was not using a standardized diagnostic interview or schedule, perhaps the authors could briefly describe process and whether any inter-rater agreement was achieved?

Response 5: Patients were not evaluated in the emergency room but when hospitalized in non-psychiatric care units during the first week after their medical condition was stabilized. The clinical evaluation was conducted over several visits and included not only individual assessment but also information from their relative(s) or significant other(s).  This was important not only for acute mental health conditions but also for others, such as PD, that need a longitudinal assessment. The diagnosis was contrasted with the psychiatrist’s in charge of the patient during the hospitalization in order to reach a high inter-rater reliability.

  • Point 6: How long did this assessment take with this vulnerable group? Were breaks allowed, how were these conducted during the week post MSSA? Was there any missing data?

Response 6: We have also described in more detail how the assessment (see Response 5). All patients completed the semi-structured interview but 30 (17.8%) did not finish the self-administered psychometric assessment. We have included this information in the Methods section.

Results

  • Point 8: Could the n be provided for those who did not have current mental health disorder (these are presumably in the no PD group)

Response 8: We have included information on patients without any mental disorders diagnosis in the Results section: 3 (0.2%) did not meet criteria for any Axis I and Axis II mental condition.

  • Point 9: As expected, there is some comorbidity in the groups, is it possible to provide range in number of multiple diagnoses? Is it possible to compare average number across groups?

Response 9: We have provided more information on the diagnostic comorbidity in the Results section.

  • Point 10: Inconsistent boldface in Table 1

Response 10: We have corrected the Table 1 accordingly.

Discussion

  • Point 11: Overall, I feel there is a need for more ‘discussion’ of these cross-sectional findings.

Response 11: We thank you for the suggestion. We have expanded that manuscript section in order to discuss the strengths and limitations of the study.

  • Point 12: As it stands, I am not fully grasping the link that “Our findings point to the need of including more complex psychological variables”. While I agree with the authors assertion, I feel the rationale for this statement needs better explaining.

Response 12: We specify what may be interesting to assess with regards to complex psychosocial variables related to suicide behaviour.

  • Point 13: The second paragraph stands out, how does this relate to current findings? Was there a prevalence of BPD in the current sample?

Response 13: Prevalence of BPD was included in the Results Section

Point 14: This section appears weak to me. “This better knowledge may help prevent the serious medical and psychological consequences of this type of suicide attempt[s] and, potentially, deaths by suicide among them.” Is this referring to the suggestions for future research or the current findings? I believe a stronger conclusion of the relevance and implications of the current study are warranted.

Response 14: We have improved this conclusion pointing to the relevance and implications of this study from the clinical and preventive point of view.

Minor comments

  • Point 15: Some editing for English sentence tense required throughout, sometimes MSSA is incorrectly abbreviated to MMSA

Response 15: We thank you for the suggestion. The new version of the manuscript has been thoroughly reviewed by a native English speaking editor.